# The Effect of Nanomaterials on DNA Methylation: A Review

**DOI:** 10.3390/nano13121880

**Published:** 2023-06-17

**Authors:** Ana Valente, Luís Vieira, Maria João Silva, Célia Ventura

**Affiliations:** 1Department of Human Genetics, National Institute of Health Doutor Ricardo Jorge, I.P. (INSA), Av. Padre Cruz, 1649-016 Lisbon, Portugal; ana.valente@insa.min-saude.pt (A.V.); luis.vieira@insa.min-saude.pt (L.V.); celia.ventura@insa.min-saude.pt (C.V.); 2Department of Animal Biology, Faculty of Sciences, University of Lisbon, Campo Grande, 1749-016 Lisbon, Portugal; 3Centre for Toxicogenomics and Human Health (ToxOmics), NOVA Medical School, NOVA University of Lisbon, 1169-056 Lisbon, Portugal

**Keywords:** nanomaterial, nanotoxicology, DNA hypomethylation, DNA hypermethylation, genome methylation, epigenetics

## Abstract

DNA methylation is an epigenetic mechanism that involves the addition of a methyl group to a cytosine residue in CpG dinucleotides, which are particularly abundant in gene promoter regions. Several studies have highlighted the role that modifications of DNA methylation may have on the adverse health effects caused by exposure to environmental toxicants. One group of xenobiotics that is increasingly present in our daily lives are nanomaterials, whose unique physicochemical properties make them interesting for a large number of industrial and biomedical applications. Their widespread use has raised concerns about human exposure, and several toxicological studies have been performed, although the studies focusing on nanomaterials’ effect on DNA methylation are still limited. The aim of this review is to investigate the possible impact of nanomaterials on DNA methylation. From the 70 studies found eligible for data analysis, the majority were in vitro, with about half using cell models related to the lungs. Among the in vivo studies, several animal models were used, but most were mice models. Only two studies were performed on human exposed populations. Global DNA methylation analyses was the most frequently applied approach. Although no trend towards hypo- or hyper-methylation could be observed, the importance of this epigenetic mechanism in the molecular response to nanomaterials is evident. Furthermore, methylation analysis of target genes and, particularly, the application of comprehensive DNA methylation analysis techniques, such as genome-wide sequencing, allowed identifying differentially methylated genes after nanomaterial exposure and affected molecular pathways, contributing to the understanding of their possible adverse health effects.

## 1. Introduction

Over the last years, there has been a great interest in understanding the mechanisms of epigenetics and their contribution to the development of human diseases [1]. Epigenetics can be defined as the study of the molecular processes that control gene expression in a potentially heritable way [1]. These mechanisms can be divided into three main categories: modifications of the histone tails, microRNA (miRNA) expression and DNA methylation [1,2].

DNA methylation is one of the most studied epigenetic mechanisms involved in gene regulation and several studies have shown that exposure to environmental toxicants can elicit changes in the DNA methylation pattern, leading to potentially adverse health effects [3,4]. One group of xenobiotics that are progressively more present in our daily lives are nanomaterials [5,6,7,8,9]. A nanomaterial is a natural, incidental or manufactured material; composed of solid particles that are present alone, in aggregates or agglomerates; and where at least 50% of these particles have external dimensions of 1 to 100 nm [10]. Nanomaterials can be organic (e.g., liposomes and dendrimers) or inorganic (e.g., titanium dioxide nanoparticles and silver nanoparticles) and are classified according to their physical (e.g., nanofibers, nanoparticles) and chemical (e.g., carbon-based, metal oxides) characteristics [11]. Due to their nano size and high surface to volume ratio (high aspect ratio), nanomaterials can have unique and interesting physicochemical properties, such as high mechanical strength, lightness and thermal and electrical characteristics [12,13]. Thus, their applications in industry (e.g., engineering, electronics, cosmetics, textiles, food additives and preservatives) and biomedicine (e.g., targeted drug delivery, regenerative medicine, diagnostic methods) are continuously increasing and, therefore, raising concerns in the context of human exposure [12,13].

To date, several in vitro and in vivo toxicological studies have shown that exposure to nanomaterials may cause adverse health effects through several biological mechanisms, including inflammation, oxidative stress, apoptosis, immunotoxicity and genotoxicity, among others [12,14,15,16,17]. Moreover, some nanomaterials, such as titanium dioxide nanoparticles (TiO_2_) or the multi-walled carbon nanotube MWCNT-7 (also known as Mitsui-7), have been classified as possible carcinogens to humans (group 2B) by the International Agency for Research on Cancer (IARC) [15]. Many of these adverse effects, such as cancer, can be related to changes in DNA methylation [15,18]. Thus, changes in DNA methylation may be one of the molecular mechanisms behind the reported toxicological effects, and by which nanomaterials affect cellular homeostasis.

DNA methylation mainly occurs on CpG dinucleotides [1,2]. In the human genome, there are regions with a high density of CpG dinucleotides, known as CpG islands, that are not normally methylated and are located close to gene promoter regions (~70% of the promoters reside within CpG islands) [2,19]. DNA methylation is mediated by a group of enzymes, known as DNA methyltransferases (DNMTs), that transfer a methyl group from S-adenosyl methionine (SAM) to the fifth carbon of a cytosine residue to form 5-methylcytosine (5-mC) [19]. Two of the DNMTs (DNMT3A and DNMT3B) are specialized in establishing new patterns of methylation (de novo methylation), while DNMT1 maintains the methylation pattern during DNA replication [19,20]. Due to methylation patterns being dynamic and responsive to stimuli, DNA demethylation can also occur passively in dividing cells or as an active mechanism in non-dividing cells through 5-mC oxidation to 5-hydroxymethylcytosine (5-hmC) by the ten-eleven translocation (TET) enzymes [19,20] (Figure 1).

DNA methylation can be analyzed using different methodologies and techniques. Some methodologies quantify the global levels of 5-mC in exposed in vivo or in vitro models, as compared with non-exposed models, allowing an overall overview of the DNA methylation levels (e.g., Enzyme-Linked Immunosorbent Assay (ELISA) [21], High performance liquid chromatography (HPLC) [22] and Liquid Chromatography with tandem mass spectrometry (LC-MS/MS) [23]), while others analyze the profile of methylation in the whole genome or in specific genes, allowing the identification of the locus with methylation differences after exposure (e.g., bisulfite sequencing [24], microarrays [25], Next Generation Sequencing (NGS) [26] and methylation-specific PCR [27]).

Nevertheless, the number of studies focusing on the effect of nanomaterials on DNA methylation is still small, either in vivo or in vitro. Knowledge of this effect could enlighten the molecular mode of action of the nanomaterials, i.e., elucidate the role of the methylation changes on gene expression, and, consequently, on the cellular pathways in which the corresponding proteins are involved and associating those changes with the expected health outcomes through adverse outcome pathways. Some studies also expect to use the specific methylation changes that are identified as biomarkers of effect in human biomonitoring [28].

Here, we provide a review of the studies that are available in the literature, with the aim of understanding the potential impact of nanomaterial exposure on DNA methylation and attempting to relate this impact with the nanomaterial’s specific physicochemical characteristics.

## 2. Materials and Methods

We performed a literature review of the existing studies on the effects of nanomaterials on DNA methylation. The search was performed on three databases of peer-reviewed literature: (1) Pubmed (all fields); (2) Web of Science (only abstracts: only included articles and toxicology category); and (3) Scopus (only title-abstract-keywords: only included articles) on 24 October 2022, using the query ((Nanomaterial* OR Nanoparticle*) AND (“DNA methylation”) AND Toxicity), with inclusion criteria of not being a review or book chapter, and being in English.

The titles and abstracts of articles were screened by two independent reviewers, and the studies were selected according to two main questions: (1) Does it include DNA methylation; (2) Is the study conducted in an animal model, either in vitro or in vivo (plants, fungi and prokaryotic organisms were excluded). Afterwards, the articles were organized and deduplicated using Excel software (version 2305, 1 June 2023). Each complete article (i.e., full text) was screened by the two reviewers independently for data extraction.

## 3. Results

The literature review retrieved a total of 181 papers from all databases (Pubmed, Web of Science and Scopus) with 110 papers screened in the first stage, following the exclusion of duplicates. In this stage, 25 studies were excluded based on not complying with the inclusion criteria (e.g., reviews, book chapters) or being non-animal models (plant and fungi). For the eligibility stage, 85 abstracts were analyzed, with 16 studies being excluded due to multiple reasons (10 were studies of biomedical applications, such as drug delivery systems, that included nanomaterials; 5 did not study the methylation of DNA, but that of other molecules, such as histones; in one study, DNA methylation changes were not identified in the same in vivo model as the one that was exposed to nanomaterials). Consequently, only 69 papers were eligible for data analysis. An overview of the results from selection workflow can be found in Figure 2.

The majority of the studies retrieved were in vitro studies (70%, 49/70), with about half using cell models related to the respiratory tract (53%, 26/49). Among the in vivo studies (30%, 21/70), the main target organs were the lungs (28.6%, 6/21) and liver (19%, 4/21). Global DNA methylation analysis was the most frequently applied approach (68.1%, 47/69), and the ELISA assay was the most used technique (30.4%, 21/69) both in vivo and in vitro. Only one article included in vitro (MCF-7 cells) and in vivo (mice) approaches, using cromolyn chitosan nanoparticles. The effect of occupational exposure to several NMs was addressed in only two studies.

Concerning the nanomaterials, TiO_2_ was the most frequently studied nanomaterial in vitro and in vivo (18.8%, 13/69), followed by silver nanoparticles (17.6%, 12/69) and carbon nanotubes (15.9%, 11/69).

The 70 studies that were found will be presented in the following tables, grouped into in vitro (Table 1), in vivo (Table 2) and occupational studies (Table 3). These tables briefly indicate which nanomaterials were studied, in which model or population, which techniques were used, and the main conclusions to be drawn with regard to DNA methylation.

## 4. Discussion

The majority of studies retrieved were performed in vitro, highlighting the still-limited knowledge about the possible effects of nanomaterial exposure on DNA methylation in human beings. Nevertheless, the relevance of the epigenetic effect of nanomaterials on disease development and progression is significant, as seen by Wen et al. [78] in mice exposed to silver nanoparticles. To date, few epidemiological studies have been conducted on this subject, with only two occupational studies identified in our search (Table 3). Rossnova and colleagues [96] used microarrays to identify the 14 most differently methylated CpG loci, and concluded that there was a significant increase and a decrease in the mean methylation on the CpG loci of the *LGR6* and *HCG27* genes, respectively, in the white blood cells of nanocomposite research workers followed during a three-year period. Both genes are associated with the development of lung carcinoma, asthma and are part of signaling pathways that could be involved in responses to exposure stressors [96]. On the other hand, Liou and colleagues studied workers from 14 manufacturing and handling nanomaterial factories exposed to TiO_2_, silica and indium tin oxide nanoparticles, and found that lipid peroxidation and DNA oxidation damage were increased in these workers, with a significant negative correlation between white blood cells 8-OHdG and global methylation only found in indium-tin-oxide-handling workers, indicating that high levels of exposure via inhalation to metal oxides may cause global hypomethylation.

In vivo studies also revealed that most nanomaterials could lead to changes in DNA methylation, although with no obvious trend towards hypo- or hyper-methylation (Table 2). These in vivo studies were mainly conducted on rodents [73,78,79,81,82,83,84,85,86,90,95] and zebrafish [80,91,92,94], but other animal models were also used, such as *Xenopus laevis* tadpoles [93] and *Enchytraeus crypticus* [87,88]. Regarding in vitro studies (Table 1), several cell lines were analyzed, mainly representative of the respiratory tract, but also of the liver, skin, intestine, immune system and others, also showing very diverse effects on DNA methylation.

The studies about global DNA methylation can provide important clues regarding nanomaterials’ adverse effects. In that regard, Gambelunghe et al. (2020) detected an increase in DNA methylation and a loss of global hydroxymethylation in pulmonary epithelial cells exposed to gold nanoparticles, which are epigenetic changes associated with cancer [38]. Thus, most studies used colorimetric or fluorometric assays that can easily measure the differences in the levels of global DNA methylation after in vivo or in vitro exposure to the nanomaterials under study [33,35,38,45,48,49,51,55,62,66,74,78,82,83,84,87,90,93,94,95]. Others used liquid chromatography with tandem mass spectrometry (LC-MS/MS), a much more powerful bioanalytical technique, to quantify global DNA methylation and hydroxymethylation [42,46,50,54,59,60,64,81,88,91,92]. Furthermore, many studies indirectly evaluated global DNA methylation through the analyses of DNA methyltransferase expression (DNMT1, DNMT2, DNMT3a and DNMT3b), a family of enzymes responsible for DNA methylation. Similarly, the study of TET expression can represent a surrogate biomarker for global DNA demethylation. Choudhury and colleagues demonstrated that ZnO-NP-induced ROS could promote global hypomethylation by triggering the expression of TET enzymes, without DNMT interferences [52]. Most studies found an interplay between both DNMTs and TETs. For instance, mice exposed to copper oxide and laser printer-emitted engineered nanoparticles showed under-expression of DNMT3a and TET1 in the alveolar macrophages and lung tissue, over-expression of DNMT1 in the alveolar macrophages and under-expression of DNMT3b in lung tissue [86]. Zhou et al. (2019) found ROS-induced methylation remodeling of zebrafish heart after 60 days of exposure to CNTs, with a dose-dependent enhancement of global genome methylation, upregulation of DNMT3b and TET2 and decreases in DNMT1, DNMT3a and TET1, associated with inflammation and cardiotoxicity [94]. Another indirect approach that evaluates global DNA methylation is the study of the methylation levels of the most frequent repetitive DNA sequences found in the human genome, since these cover a large portion of the genome sequence. For instance, Alu and LINE are, respectively, short and long interspersed repeats found broadly distributed throughout the genome, either in transcriptionally inactive or active loci (that correspond to SINE B2 and SINE B1 in mice). Thus, the epigenetic status of these two types of repeats may be used as proxies for the evaluation of the global levels of DNA methylation across the whole genome, and some studies have applied this approach [44,50,52,56,65,76,86,95]. Other repeats can also be used to evaluate specific regions, e.g., subtelomeric D4Z4 repeat or the centromeric SATα repeats [52,65]. In the two studies analyzing specific repeats, their methylation profile was in line with the overall results, i.e., no change [52] or hypomethylation [65].

Nevertheless, there can exist important differences in the methylation patterns between different loci that are not detected using global DNA methylation approaches, since simultaneous hypomethylation of some loci and hypermethylation of others may result in cumulatively undetectable changes in the levels of DNA methylation. Accordingly, although no change in the overall DNA methylation was found in A549 cells exposed to TiO_2_ for 4 or 24 h, there was a moderate increase in the promoter methylation of DNA repair genes [42]. Moreover, mice exposed to gold nanoparticles (AuNP) and carbon nanotubes (CNT) showed no effect on lung global methylation and hydroxymethylation, but AuNP induced methylation changes in Atm, Cdk, Gsr, Gpx and Trp53 genes, and SWCNT induced promoter hypomethylation in the Atm gene [81]. Additionally, Zhou et al. (2019) found global hypermethylation in zebrafish hearts after being exposed to CNTs, but the CpG dinucleotide sites at lepb, cd248b and il11a gene promoters, which are genes associated with inflammation and hemostasis, showed decreased methylation [94]. Thus, global methylation analysis may not detect relevant biological methylation changes, since one of its limitations is the inability to identify the genes that change their methylation profile. Therefore, other researchers focused on the analysis of the methylation of target genes related to the outcomes of interest [81]. Increased promoter methylation of TNF-α and Thy-1 genes, associated with inflammation and fibrosis [83], and Hif1α activity, associated with a variety of tumors and oncogenic pathways [97], were identified after exposure to TiO_2_ [82]. Zhang and colleagues [79] showed that silver nanoparticles altered the methylation of the imprinted gene Zac1 in placentas of pregnant mice, and Xu and colleagues [80] concluded that myogenic loci-specific DNA methylation resulted in muscle dysfunction in zebrafish embryos.

With the development of high-throughput genomic methodologies, as genomic arrays or next-generation sequencing, an “omics” approach became possible in toxicology [98]. Thus, some authors applied these methodologies to study genome-wide methylation, revealing extensive changes covering hundreds or even thousands of different hyper- or hypo-methylated CpGs [47,57,59,60,68]. With this approach, it was possible to identify which biological or molecular pathways were being impacted by the exposure to nanomaterials. For instance, the mitochondrial-mediated apoptosis of epithelial bronchial cells exposed to silica nanoparticles was linked to the downregulation of the PI3K/Akt/CREB/Bcl-2 signaling pathway [69]. Wang et al. (2021) found that the differently methylated genes in BEAS-2B cells exposed to SWCNT were associated with nucleotide-excision repair, cell differentiation, extracellular matrix organization, cell junction and other cancer-related processes, as well as with insulin resistance and the AMPK and mTOR signaling pathways [57]. The genome-wide DNA methylation changes caused by silica nanoparticles in GC-2 cells highlighted the possible influence of silica nanoparticles in male reproductive toxicity [68], and in lipid metabolism disorders and cancer, through p53-mediated apoptotic pathway inhibition in mouse liver and activation of the HRAS-mediated MAPK signaling pathway [95].

From the global analyses of all retrieved studies, it was not possible to identify a general trend on the DNA methylation effects of nanomaterials. Even for the same type of nanomaterial, different consequences in DNA methylation may be observed, depending on its specific physicochemical characteristics. For instance, Scala et al. (2018) linked the patterns of genomic and epigenomic regulation to the intrinsic properties of the carbon nanomaterials under study (particle and spherical materials, carbon black and fullerene, and tubes and fibers) [63]. Chatterjee et al. (2017) also observed differences in global DNA methylation between hydroxylated and carboxylated MWCNTs [62]. Oner et al. (2020) showed that SWCNTs, but not MWCNTs, induced hypo- or hypermethylation on CpG sites in DNA after a very low-dose exposure and recovery period [58]. Hu et al. (2019) also found that the chemical groups in the surface of graphene quantum dots were a critical factor for modulating DNA methylation [92] and Ma et al. (2017) showed that only the anatase TiO_2_ caused differences in DNA methylation [45].

Moreover, different nanomaterial concentrations resulted in different methylation effects, with some studies showing a dose-response effect [61,71]. Also, different cell types responded differently to the same nanomaterials [63], and some studies suggested a dynamic molecular adaptation to nanomaterial exposure, with different methylation patterns over time and a contribution of DNA methylation in the long-term adaptation [44,46,61,92,95]. Thus, the lack of reproducible results in the different studies can be a consequence of the lack of uniformity of the methodology applied, and also of the sensitivity of the methods used.

This review did not intend to be systematic, but rather to summarize the state-of-the-art knowledge about the potential effects of nanomaterials on DNA methylation. Although all articles considered in this review presented sounded data obtained using well-described methods/tools, further refinement of the quality of toxicological data can be achieved using software-based tools, such as “ToxRTool” (Toxicological data Reliability Assessment Tool) [99]. Nevertheless, it is evident that studies on the effects of nanomaterials at the level of methylation of specific genes are of great value in discovering the adverse effects that these nanomaterials may have, for example, inflammatory, immunotoxic or carcinogenic effects, by allowing us to identify which are the molecular pathways that underlie these adverse effects. Furthermore, by using comprehensive DNA methylation analysis techniques, such as global genome sequencing, these studies will not only identify target genes and molecular pathways already suspected to be involved in the observed adverse effects, but also identify other new changed genes and pathways with an exploratory and innovative role.

## 5. Conclusions

In conclusion, the study of the effects of nanomaterials on DNA methylation is still a limited area of research. Nevertheless, although some studies revealed no effects, the majority evidenced that there was an effect at the genome level, and some identified relevant methylation changes in specific genes, highlighting the importance of considering epigenomics in the understanding of the molecular mechanisms that lead to the adverse effects caused by nanomaterial exposure. This understanding is of the utmost importance, considering the increasing role that nanomaterials play in our daily lives.

## Figures and Tables

**Figure 1 nanomaterials-13-01880-f001:**
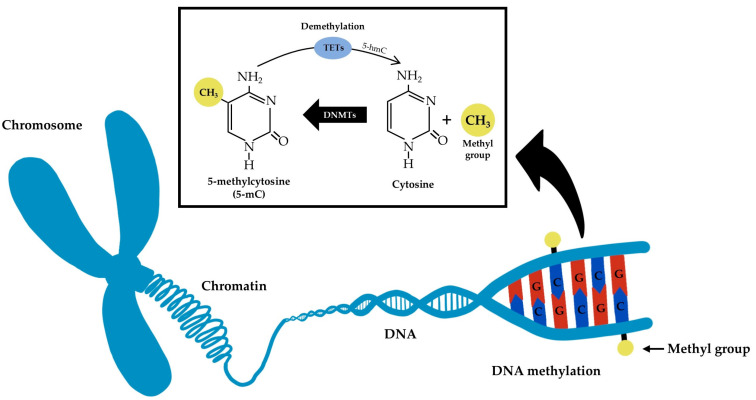
Simplified schematic representation of DNA methylation and demethylation molecular mechanisms.

**Figure 2 nanomaterials-13-01880-f002:**
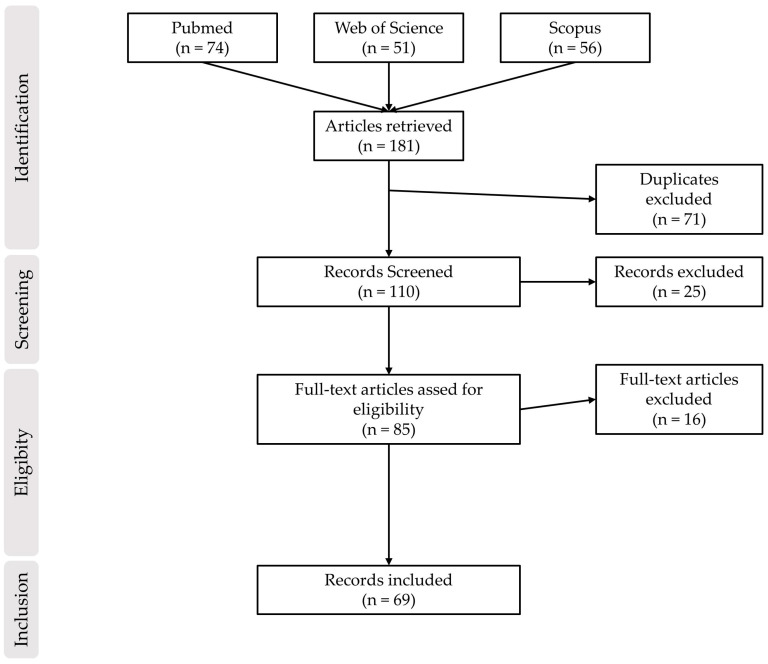
Workflow and results of the literature screening.

**Table 1 nanomaterials-13-01880-t001:** List of the in vitro studies found in the literature search, presented by type of nanomaterial.

Nanomaterials, Concentration-Range	Cell Model	Methodology	Main Conclusions	References
**Metal-based nanomaterials**				
AgNPs, 10–200 µg/mL for 72 h.	A549 cells	Global DNA methylation (ELISA)	High concentrations of AgNPs for 72 h induced higher levels of global DNA methylation.	Blanco et al., 2017 [29]
AgNPs with a diameter of 10, 50 or 100 nm (nAg10, nAg50 and nAg100, respectively), 10 μg/mL for 24 h.	A549 cells	Global DNA methylation (immunofluorescence staining); DNMT1 and 3B expression (Western blot)	All AgNPs decreased DNA methylation, decreased DNMT1 and increased DNMT3B. nAg10 may induce DNA hypomethylation through a proteasome-mediated degradation of DNMT1.	Maki et al., 2020 [30]
AgNPs, 1 µg/mL.	BEAS-2B	Genome-wide methylation (array)	Only 1 differentially methylated promoter of a poorly characterized gene, 6 differentially methylated CpG sites and 5 differentially methylated tiling regions located in intergenic regions with regulatory function.	Gliga et al., 2018 [31]
AgNPs capped with glutathione, NA.	HepG2 cells	mRNA and microRNA expression (RNAseq)	DNA methylation was an affected pathway identified in the bioinformatics pathway analyses.	Thai et al., 2021 [32]
AgNPs, 5 μg/mL for 48 h, with or without 96 h without treatment.	HT22 cells	Global DNA methylation (ELISA), DNMTs expression (ELISA)	Increase in 5-mC, DNMT1, DNMT2, DNMT3a and DNMT3b levels. The upregulation of DNMT2 may be a part of cellular stress response to AgNPs.	Mytych et al., 2017 [33]
AgNPs, 1–20 μg/mL for 24 h.	HEK293T cells	Genome-wide methylation (MeDIP-Seq)	A total of 12 up-regulated and hypomethylated genes and 22 down-regulated and hypermethylated genes, which were primarily involved in lipid and ion metabolism.	Chen et al., 2022 [34]
AgNPs, 0.1–100 μM for 3, 6, 12 and 24 h.	EA.hy926 cells	Global DNA methylation (flow cytometry)	Increased global DNA methylation.	González-Palomo et al., 2021 [35]
AgNPs, 5–25 μg/mL for 24 h.	NIH3T3 cells	Transcriptomics (RNAseq)	Alterations in epigenetic-related processes including nucleosome assembly and DNA methylation.	Gurunathan et al., 2018 [36]
AgNPs (10 µg/mL), AuNPs (10 µg/mL) and superparamagnetic iron oxide nanoparticles (5 µg/mL) for 24 h.	HepG2 cells	Promoter methylation of genes related to inflammatory response and apoptosis	No differences in the methylation for any type of nanoparticle.	Brzóska et al., 2019 [37]
AuNPs, 0.8 and 1.6 µg/cm^2^ for 3, 24 and 48 h.	BEAS-2B and A549 cells	Global DNA methylation (ELISA)	LPS-challenged cells underwent an increase in global DNA 5-mC and a decrease in global 5-hmC, possibly associated with a carcinogenic-like transformation process.	Gambelunghe et al., 2020 [38]
AuNPs, 1 nM for 48 or 72 h.	MRC-5 cells	Methylation of the PROS1 gene	No changes in PROS1 methylation.	Ng et al., 2011 [39]
AuNPs, 10, 50 and 100 μg/mL for 72 h.	HEK293 and HaCaT cells	Global DNA methylation (Immunocytochemical staining)	No differences in methylation in HEK293 cells and HaCaT cells treated with AuNPs.	Sooklert et al., 2019 [40]
AuNPs of 1.5, 4 and 14 nm core sizes, 10 μg/mL for 24 h.	hESCs	Genome-wide methylation (array)	Thiolate-capped 4 nm AuNPs at 10 g/mL caused a dramatic decrease in global DNA methylation and an increase in global DNA hydroxymethylation in only 24 h.	Senut et al., 2016 [41]
TiO_2_, 1–100 μg/mL for 4, 24 and 48 h.	A549 cells	Global DNA methylation (HPLC-MS/MS) and methylation profile of 20 DNA repair gene promoters (qPCR)	No change in the overall DNA methylation. Exposure to 100 µg/mL TiO_2_ for 4 or 24 h led to a moderate increase in APE1, POLD3, MRE11A and PMS2 methylation.	Biola-Clier et al., 2017 [42]
TiO_2_, 6.25–100 μg/mL for 24 h.	A549 cells	PARP-1 methylation	Increased levels of PARP-1 methylation.	Bai et al., 2015 [43]
TiO_2_ uncoated and coated with silica or citrate, 40 μg/cm^2^ for 48 and 72 h.	A549 cells	LINE-1 methylation (ELISA)	No effect on LINE-1 methylation after 48 h. After 72 h, a reduction in global DNA methylation levels was induced by all nanoparticles.	Stoccoro et al., 2017 [44]
TiO_2_, 0.1–100 μg/mL^−1^ for 48 h.	16HBE and A549 cells	Global DNA methylation (ELISA) and DNMT3B and TET expression (qPCR)	Only the anatase-type TiO_2_ decreased global DNA methylation and altered expression levels of methylation-related genes and proteins.	Ma et al., 2017 [45]
TiO_2_, 3.25, 12.5 and 25 μg/mL for 3 and 24 h.	16HBE cells	Global DNA methylation (LC-MS/MS)	Increased global DNA methylation at 24 h for anatase and rutile TiO_2_, and the mixture of both.	Ghosh et al., 2017 [46]
TiO_2_, 10 and 20 mg/mL for up to 4 weeks.	BEAS-2B cells	Genome-wide methylation	In total, 755 CpG sites were identified with only minor consistent effects of hypomethylation.	Sierra et al., 2017 [47]
TiO_2_, 100 μg/mL for 24 or 72 h.	Caco-2, HepG2, NL20 and A-431 cells	Global DNA methylation (ELISA), gene-specific methylation (array) and expression of DNMTs, MBD2 and UHRF1	Decrease in global DNA methylation in Caco-2, HepG2 and A-431 cells. Across the four cell lines, eight genes (CDKN1A, DNAJC15, GADD45A, GDF15, INSIG1, SCARA3, TP53 and BNIP3) with promotors methylated after exposure. Altered expression of DNMT1, DNMT3A, DNMT3B, MBD2 and UHRF1, which was cell-type-dependent.	Pogribna et al., 2020 [48]
TiO_2_, 25–100 μg/mL for 24 h.	PBMCs	Global DNA methylation (ELISA)	DNA hypomethylation.	Malakootian et al., 2021 [49]
TiO_2_ and CuONPs, 0.5 and 30 µg/mL for 24 h.	THP-1, RAW264.7 and SAEC cells	Global DNA methylation (LC-MS/MS); LINE1 and Alu/SINE element methylation (methylation-sensitive qPCR) and expression of DNA methylation machinery (qPCR)	Modest alterations in methylation of LINE-1 and Alu/SINE, and decreased expression of DNA methylation machinery in a cell-, dose- and nanomaterial-dependent manner.	Lu et al., 2016 [50]
TiO_2_ and ZnO, 0.125–8 μg/mL for 24, 48 and 72 h.	MRC-5 cells	Global DNA methylation (ELISA), DNMTs activity (ELISA) and expression (qPCR)	Decrease in global DNA methylation and DNA methyltransferase activity. Direct correlation between nanoparticle concentration, global DNA methylation and expression of DNMT1, 3A and 3B.	Patil et al., 2016 [51]
ZnO, 25 and 50 μg/mL for 48 h.	HEK-293 cells	Global and locus-specific DNA-methylation at LINE-1, D4Z4 and NBL2, and at the promoter of selected ROS-responsive genes (AOX1, HMOX1, NCF2, SOD3). Global DNA methylation, DNMTs and TET activity	Global reduction in 5-methylcytosine and increase in 5-hydroxymethylcytosine. Significant increase in the expression of TETs, but not in the expression of DNMTs.	Choudhury et al., 2017 [52]
CuNPs, NA.	HepG2 cells	mRNA and microRNA expression (RNAseq)	mRNA–microRNA interaction revealed altered DNA methylation. Altered expression of DNMTs.	Thai et al., 2021 [53]
CuONPs, 6.25 to 400 mg L^−1^ for 24 h.	N2A cells	Global DNA methylation (HPLC)	Changes in the m5dC/dC ratio were less than 1%, which may indicate that CuONPs do not alter DNA methylation in vitro.	Perreault et al., 2012 [54]
Maghemite nanoparticles covered with citric acid at 0.5 and 3.0 mg FemL^−1^ for 24 or 48 h.	HSG cells	Global DNA methylation (ELISA)	Altered global DNA methylation with reduced expression of genes related to epigenetic reprogramming.	Bonadio et al., 2020 [55]
Pristine plasma and laser ITER-like tungsten nanoparticles, 1–5 µg/mL for 24 h.	BEAS-2B cells	DNA methylation of Alu, LINE and Satellite 2 and 3 (Sat 2 and Sat 3) repeats (bisulfite sequencing)	No significant changes in DNA methylation.	Uboldi et al., 2019 [56]
**Carbon-based nanomaterials**				
SWCNTs, 10 µg/mL with 30- and 60-day recovery periods.	BEAS-2B cells	Genome-wide methylation (array); gene-specific methylation (methylation-specific PCR)	DNMT3A and DNMT1 up-regulation after 30–60 days of recovery. A total of 457 hypermethylated and 367 hypomethylated gene promoters. Hypermethylation of PIM2 gene and hypomethylation of ABCA2 and CRYBG3 genes in the 60-day recovery period group.	Wang et al., 2021 [57]
MWCNTs or SWCNTs, 0.25 µg/mL for four weeks and recovery period of two weeks.	16HBE cells	Genome-wide methylation (array)	MWCNTs induced a single hypomethylation at a CpG site on a gene promoter. Exposure to SWCNTs induced hypermethylation at CpG sites which may involve ‘transcription factor activity’ and ‘sequence-specific DNA binding’ gene ontologies. After the recovery period, no change in DNA methylation for MWCNTs, and hypermethylation and hypomethylation for SWCNTs. HPCAL1, PRSS3, KLK3, KLF3 genes were hypermethylated at different time points in SWCNT-exposed cells.	Öner et al., 2020 [58]
MWCNTs and SWCNTs, 25 and 100 μg/mL for 24 h.	16HBE cells	Global DNA methylation and hydroxymethylation (LC-MS/MS); whole-genome methylation (array)	MWCNTs hypomethylated 2398 gene promoters; after exposure to SWCNTs, 589 CpG sites (located on 501 genes) were either hypo- (N = 493) or hypermethylated (N = 96). Differentially methylated and expressed genes induced changes (MWCNTs > SWCNTs) at different cellular pathways, such as p53 signaling, DNA damage repair and cell cycle. SWCNT exposure showed hypermethylation on SKI, GTSP1, SHROOM2 and NF1 genes.	Öner et al., 2018 [59]
MWCNTs and SWCNTs, 25 and 100 μg/mL for 3 and 24 h.	THP-1 cells	Global DNA methylation (LC-MS/MS), genome-wide CpG site-specific methylation (array)	No global DNA 5-mC or 5-hmC changes. MWCNTs hypomethylated 3340 promoter regions (2398 genes), with no differential methylation at individual CpG sites. SWCNTs hypomethylated 5 gene promoters (AKAP8L, FOXK2, EIF4E, snoU13 and RP11-223 l10.1); 493 hypomethylated and 96 hypermethylated single CpG sites, located on 501 different genes.	Öner et al., 2017 [60]
MWCNTs, 5, 10 and 15 μg/mL for 24, 48 and 72 h.	THP-1 cells	Genome-wide methylation (array)	Increasing dose-dependent trend of differentially methylated promoters at 24 h and a dose-dependent decrease in hypomethylated promoters at 48 h.	Saarimäki et al., 2020 [61]
MWCNTs, 10 and 20 mg/mL for up to 4 weeks.	BEAS-2B cells	Genome-wide methylation	755 CpG sites were identified with only minor consistent effects of hypomethylation.	Sierra et al., 2017 [47]
Functionalized MWCNTs (hydroxylation (8.37 and 6.34 mg/L); carboxylation (37.99 and 4.44 mg/L) and pristine (2.92 and 2.17 mg/L))	BEAS-2B and HepG2 cells	Global DNA methylation (ELISA)	DNMT3B-dependent hypo-methylation in BEAS-2B cells and hyper-methylation in HepG2 cells in a functionalization-dependent manner.	Chatterjee et al., 2017 [62]
Carbon black, fullerene, graphite nanofibers, SWCNTs and MWCNTs, 0.1–500 μg/mL for 48 h.	A549, BEAS-2B and THP-1 cells	Genome-wide methylation (array)	Molecular alterations are highly dependent on the cell type and geometrical properties of the carbon nanomaterials.	Scala et al., 2018 [63]
Fullerene, long or short MWCNTs or SWCNTs, 0.1 mg/L.	A549 cells	Global DNA methylation (HPLC-MS), DNMTs expression (qPCR)	Increased global DNA methylation. Down-regulating tendency in DNMT transcription, except for C60, but only significant for DNMT3b after SWCNT treatment.	Li et al., 2016 [64]
GONPs, 1 and 10 µg/mL for 15 and 30 days.	BEAS-2B cells	LINE-1, D4Z4 and NBL2, SATα and AluYb8 methylation (bisulfite pyrosequencing); genome-wide DNA methylation (array)	No genome-wide or global DNA methylation changes associated with either condition.	Pérez et al., 2020 [65]
Pristine, carboxylated and aminated graphene, graphene nanoplatelets, SLGO and FLGO, 10 and 50 mg/L for 24 h.	BEAS-2B cells	Global DNA methylation (ELISA)	Increased global DNA methylation after exposure to SLGO/FLGO and decreased global DNA methylation after exposure to the remaining nanoparticles.	Chatterjee et al., 2016 [66]
GQD, 50 μg/mL.	mESCs	DNMT1, DNMT3A, DNMT3B, TET1, TET2 and TET3 gene expression (qPCR); methylation of Sox2 and Oct4 promoter regions (bisulfite treatment and NGS)	GQD-induced inhibition in CpG methylation of Sox2 through altering methyltransferase and demethyltransferase expression.	Ku et al., 2021 [67]
**Silica nanoparticles**				
SiNPs, 20 μg/mL for 24 h.	GC-2 cells	MeDIP-seq	Extensive methylation changes, with a total of 428 hyper-methylated genes and 398 hypo-methylated genes, probably involved with abnormal transcription and translation, mitochondrial damage and cell apoptosis.	Sang et al., 2021 [68]
SiNPs, 3.125–100 μg/mL for 24 h.	BEAS-2B cells	Genome-wide methylation (array)	Of the 25 significant altered processes, the apoptosis-related PI3K/Akt pathway involved 32 differentially methylated gene promoters, in which CREB3L1 and Bcl-2 were hypermethylated, in association with the downregulation of their mRNA levels.	Zou et al., 2016 [69]
SiNPs, 10, 50 and 100 μg/mL for 72 h.	HEK293 and HaCaT cells	Global DNA methylation (Immunocytochemical staining)	No differences in methylation in HEK293 cells, but HaCaT cells exposed to 10 µg/mL SiNPs had lower levels of methylation.	Sooklert et al., 2019 [40]
SiO_2_NPs, 2.5, 5 and 10 μg/mL for 24 h.	HaCaT cells	Promoter methylation of PARP-1 (methylation-specific PCR and bisulfite sequencing)	Decrease in PARP-1 mRNA and protein levels and a simultaneous increase in PARP-1 methylation.	Gong et al., 2012 [70]
SiNPs, 2–10 µg/mL for 48 h.	HaCaT cells	Global DNA methylation (flow cytometry); DNMTs expression (qPCR and Western blot)	Decreased levels of DNMT1, DNMT3A and MBD2 in a dose-dependent manner at mRNA and protein levels. Global DNA methylation decreased with dose.	Gong et al., 2010 [71]
SiNPs, 2 and 5 μg/cm^2^ for 6 days.	Bhas 42 cells	Global DNA methylation (ELISA) and DNMTs expression (Western blot)	SiNPs treatment did not affect DNMT3A and DNMT3B expression or DNA methylation.	Seidel et al., 2017 [72]
**Other nanomaterials**				
CSNPs, 10, 50 and 100 μg/mL for 72 h.	HEK293 and HaCaT cells	Global DNA methylation (Immunocytochemical staining)	No differences in methylation in HEK293 cells, but HaCaT cells exposed to 100 µg/mL CSNPs had lower levels of methylation.	Sooklert et al., 2019 [40]
Cromolyn CSNPs and CSNPs, 62.5, 125, 250, 500 μg/mL for 48 h.	MCF-7 cells	Global DNA methylation (DNMT1 and METTL3 expression (qPCR); methylation of RASSF1A and p16 genes (methylation-specific PCR)	Reduction in DNMT1 expression, reversed hypermethylation pattern of RASSF1A and p16 genes and lower expression of METTL3. Cromolyn chitosan nanoparticles may act by inhibiting ERK1/2 phosphorylation/DNMT1/DNA methylation, possibly impacting RNA methylation machinery via METTL3 expression.	Motawi et al., 2022 [73]
ChiNH/Q, 10–1000 μg/mL for 48 h.	HepG2 cells	Global DNA methylation (ELISA) and expression of DNMTs (qPCR)	Reduced expression levels of DNMT1/3A/3B and increased levels of 5-mC.	Abbaszadeh et al., 2020 [74]
Dendrosomal nano curcumin, 0–45 μM for 24 h and 0–38 μM for 48 h.	HepG2 and Huh7 cells	DNMT1, DNMT3A and 3B expression (semi-quantitative and qPCR)	Downregulation of DNMT1, DNMT3A and DNMT3B expression in both cell lines.	Chamani et al., 2016 [75]
PEPs, 0.5–100 μg/mL for 8 h or more.	SAECs, THP-1 and TK6 cells	Methylation and expression of transposable elements (TEs) (qPCR), LINE-1 copy number analysis (qPCR), expression of DNMT1, DNMT3A, DNMT3b, UHRF1 and TET1, TET2, TET3 (qPCR)	Dysfunction of the DNA methylation and demethylation machinery associated with the loss of DNA methylation and the reactivation of TEs.	Pirela et al., 2016 [76]
PEPs and mild steel welding fumes, 0.5 and 30 µg/mL for 24 h.	THP-1, RAW264.7 and SAEC cells	Global DNA methylation (LC-MS/MS); LINE1 and Alu/SINE element methylation (methylation-sensitive qPCR) and expression of DNA methylation machinery (qPCR)	Modest alterations in methylation of LINE-1 and Alu/SINE, and decreased expression of DNA methylation machinery in a cell-, dose- and nanomaterial-dependent manner.	Lu et al., 2016 [50]
GO-AgNPs, 4 and 8 μg/mL for 24 h.	CFFCs cells	Global DNA methylation, DNMTs expression (qPCR)	DNA hypomethylation and decreased expression of DNMT3A.	Yuan et al., 2021 [77]

AgNPs: Silver nanoparticles; RNAseq: RNA sequencing; MeDIP-Seq: Methylated DNA immunoprecipitation sequencing; NA: Not available; AuNPs: Gold nanoparticles; LPS: Lipopolysaccharide; TiO_2_: Titanium Dioxide; qPCR: Quantitative polymerase chain reaction; LINE: Long interspersed nuclear elements; SINE: Short interspersed nuclear elements; DNMTs: DNA methyltransferases; PBMCs: Peripheral Blood Mononuclear Cells; ZnO: Zinc oxide; CuNPs: Copper nanoparticles; CuONPs: Copper oxide nanoparticles; HPLC: High-performance liquid chromatography; SWCNTs: Single-walled carbon nanotubes; MWCNT: Multi-walled carbon nanotubes; HPLC-MS: High-Performance Liquid Chromatography-Mass Spectrometry; GONPs: Graphene oxide nanoparticles; SLGO: Single layer graphene oxide; FLGO: Few-layer graphene oxide; GQD: Graphene-based quantum dots; SiNPs: Silica nanoparticles; SiO_2_NPs: Silica oxide nanoparticles; CSNPs: Chitosan nanoparticles; ChiNH/Q: Chitosan-based quercetin nanohydrogel; PEPs: Printer-emitted engineered nanoparticles; GO-AgNPs: Graphene oxide-silver nanoparticles.

**Table 2 nanomaterials-13-01880-t002:** List of the in vivo studies found in the literature search, presented by type of nanomaterial.

Nanomaterials, Exposure Conditions	Animal Model	Methodology	Main Conclusions	References
**Metal-based nanomaterials**				
AgNPs, 0.5, 2.5 and 12.5 mg/kg BW for 7 days.	C57BL/6J mice	Global DNA methylation (ELISA)	Decreased global DNA methylation and DNA hydroxymethylation in the livers of mice with high-fat-diet-induced non-alcoholic fatty liver disease (NAFLD), contributing to NAFLD development and progression.	Wen et al., 2022 [78]
AgNPs, 1.0 mg/kg for 17.5 days.	ICR mice	Methylation of Zac1 and Igf2r genes (bisulfite sequencing)	AgNP exposure significantly altered the methylation levels of Zac1.	Zhang et al., 2015 [79]
AgNPs, 0.4 mg/L.	Zebrafish embryos	Gene-specific methylation	Myogenic loci-specific DNA methylation might result in muscle dysfunctions in treated embryos.	Xu et al., 2018 [80]
AuNPs of 5, 60 and 250 nm, 2.5 mg/kg and 0.25 mg/kg for 48 h.	BALB/c mice	Global DNA methylation and hydroxymethylation (LC-MS), gene-specific methylation of 17 genes (bisulfite pyrosequencing)	AuNP exposure had no effect on 5-mC and 5-hmC levels in mouse lungs. AuNP 60 nm induced CpG hypermethylation in Atm, Cdk and Gsr genes and hypomethylation in Gpx; Gsr and Trp53 showed changes in methylation between low- and high-dose AuNP, 60 and 250 nm, respectively, and AuNP had size effects on methylation for Trp53.	Tabish et al., 2017 [81]
TiO_2,_ 12 mg/mL for 6 h/day for 6 non-consecutive days.	FVB/NJ pregnant dams and fetal pups	Global DNA methylation (ELISA), Hif1α activity (ELISA), DNMT activity (colorimetric assay)	DNA methylation was significantly increased in fetal pups following maternal exposure, along with increased Hif1α activity and DNMT1 protein expression.	Kunovac et al., 2019 [82]
TiO_2_ of 25 nm, 80 mg/mL for 30 days.	NIH mice	Global DNA methylation (ELISA), promoter methylation of IFN-gama, TNF-alfa, Thy-1	Decreased global DNA methylation and hydroxymethylation in the lung tissue only in the young group. Altered methylation of TNF-alfa and Thy-1 promoters with a role in inflammation and fibrosis.	Ma et al., 2019 [83]
CuNPs, 6.5 and 3.25 mg/kg for 4 weeks.	Wistar rats	Global DNA methylation (ELISA)	Lowering the level of copper nanoparticles in the diet increased DNA methylation.	Ognik et al., 2019 [84]
CuONPs, 3.3 mg m^−3^ and 13.2 mg m^−3^ for 6 h.	Wistar Unilever outbred rats	Methylation in inflammation-related genes (PCR array coupled with DNA restriction kit)	No aberrant DNA methylation of inflammation-associated genes.	Costa et al., 2018 [85]
CuONPs, 2.5 mg/kg body weight.	BALB/c mice	Global DNA methylation (LC-MS/MS), methylation at the LINE-1 and SINE B1 elements (methylation-sensitive qPCR), DNMT1/3A/3B expression (qPCR)	CuONPs increased the 5-mC and 5-hmC levels in lung tissue. CuONPs reduced the expressions of DNMT1, 3a and 3b in the lung tissue, but not in alveolar macrophages. The expression of TET1 decreased in both alveolar macrophages and lung tissue after exposure to CuO.	Lu et al., 2016 [86]
CuONPs, 0–500–1400 mg Cu/kg soil (DW).	*Enchytraeus crypticus* (soil invertebrate)	Global DNA methylation (immunostaining)	Differences in 5-mC between *E. crypticus* generations after exposure.	Bicho et al., 2021 [87]
CuONPs, 500–1400 mg Cu/kg soil for 32 days.	*Enchytraeus crypticus*	Global DNA methylation (LC-MS), gene-specific DNA methylation (qPCR and MS-HRM) and bisulfite sequencing	Multigenerational long-term exposure to CuO NMs induced changes in epigenetic markers. However, global DNA methylation and gene-specific methylation did not confirm the epigenetic effect.	Bicho et al., 2020 [88]
Nickel oxide nanoparticles, NA.	Mice with pulmonary fibrosis	Genome-wide methylation (whole-genome bisulfite sequencing) and transcriptomics	Hypomethylation in lung fibrotic tissue. mRNA transcriptome data found 93 DNA methylation genes with transcriptional significance.	Zheng et al., 2022 [89]
Metal-rich welding nanoparticles, 2.0 mg/rat for 30 days.	Sprague-Dawley rats	Global DNA methylation (ELISA)	No significant differences were observed when comparing DNA methylation between the welding fume and control groups at any of the time points.	Shoeb et al., 2017 [90]
**Carbon-based nanoparticles**				
SWCNTs and MWCNTs, 2.5 mg/kg and 0.25 mg/kg for 48 h.	BALB/c mice	Global DNA methylation and hydroxymethylation (LC-MS), gene-specific methylation of 17 genes (bisulfite pyrosequencing)	SWCNT and MWCNT exposure had no effect on 5-mC and 5-hmC levels in mouse lungs. SWCNT exposure induced promoter hypomethylation in Atm.	Tabish et al., 2017 [81]
SWCNTs, MWCNTs and fullerene, 0.1 mg/L for 21 days.	Zebrafish	Global DNA methylation (LC-MS)	Increased global genomic methylation, most profound in female zebrafish brain tissues, after exposure to short MWCNTs or SWCNTs.	Gorrochategui et al., 2017 [91]
GQD, 2, 10 and 50 mg/L for 7 days.	Zebrafish	Global DNA methylation (LC-MS/MS)	Global DNA hypermethylation in various tissues in a dose-dependent manner. The global DNA methylation of reduced and aminated GQD exposure increased in intestines even at low concentrations. Fourteen days after exposure, the effects had ceased. DNA methylation in the livers of fish from exposure groups was higher, even after exposure had ceased, indicating a more complex mechanism of DNA methylation deregulation.	Hu et al., 2019 [92]
GONPs with cis-bifenthrin (cis-BF) (0.06 and 0.3 μg/L) or cis-BF alone (0.1 mg/L) for 21 days.	*Xenopus laevis* (tadpoles)	Global DNA methylation (ELISA)	Reduced levels of genomic DNA methylation were observed in the co-exposure groups.	Li et al., 2020 [93]
Black carbon, 10 and 30 μg/mL for 60 days.	Zebrafish	Global DNA methylation (ELISA), promoter methylation (touch-down PCR of bisulfite-treated DNA)	Increased global genome methylation in a dose-dependent manner, upregulation of the mRNA content of DNMT3B and TET2 in heart tissue and dose-dependent decreases in the mRNA expression of DNMT1, DNMT3A and TET1. Increased unmethylated CpG dinucleotide sites at lepb, cd248b and il11a promoters.	Zhou et al., 2019 [94]
**Other nanoparticles**				
Food-grade precipitated silica (S200) and fumed silica Aerosil 200F (A200F), 225, 1000 and 5000 mg/kg for 28 and 84 days.	BALB/c mice	Global DNA methylation (ELISA), methylation levels of LINE-1 and SINEB1 (pyrosequencing), genome-wide methylation (NGS)	Changes in whole-genome methylation in peripheral mouse leukocytes and liver after 28 days. After 84 days of high-dose continuous exposure, differential methylation was mainly found in introns, intergenic regions and promoters.	Lu et al., 2021 [95]
PEPs, 2.5 mg/kg body weight.	BALB/c mice	Global DNA methylation (LC-MS/MS), methylation at the LINE-1 and SINE B1 elements (methylation-sensitive qPCR), DNMT1/3A/3B expression (qPCR)	PEPs increased the 5-mC and 5-hmC levels in lung tissue. Hypermethylation of the LINE-1 element was observed in mouse lung tissue after exposure to PEPs. PEPs increased the expression of DNMT1 in the alveolar macrophages and down-regulated DNMT3a expression in the alveolar macrophages and lung tissue; the expression of TET1 decreased in both alveolar macrophages and lung tissue after exposure.	Lu et al., 2016 [86]
CSNPs, 5 mg cromolyn/kg twice a week for 2 weeks.	Swiss albino mice injected with Ehrlich ascites carcinoma cells subcutaneously	DNMT1 and METTL3 expression (qPCR)	Cromolyn CSNPs lessened the tumor volume and halted DNMT1 and METTL3 expression in Ehrlich carcinoma mice.	Motawi et al., 2022 [73]

AgNPs: Silver nanoparticles; AuNPs: Gold nanoparticles; LC-MS: Liquid chromatography-mass spectrometry; TiO_2_: Titanium Dioxide; qPCR: Quantitative polymerase chain reaction; DNMTs: DNA methyltransferases; LINE: Long interspersed nuclear elements; SINE: Short interspersed nuclear elements; CuNPs: Copper nanoparticles; CuONPs: Copper oxide nanoparticles; MS-HRM: Methylation-Sensitive High Resolution Melting; NA: Not available; HPLC: High-performance liquid chromatography; SWCNTs: Single-walled carbon nanotubes; MWCNTs: Multi-walled carbon nanotubes; GONPs: Graphene oxide nanoparticles; CSNPs: Chitosan nanoparticles; PEPs: Printer-emitted engineered nanoparticles.

**Table 3 nanomaterials-13-01880-t003:** List of occupational studies found in the literature search, presented by type of nanomaterial.

Nanomaterial	Population	Methodology	Main Conclusions	References
PM < 25–100 nm	14 nanocomposite research workers (10 exposed and 4 controls) from 2016 to 2019.	Genome-wide methylation (microarray)	Shift in individual DNA methylation patterns in the blood of all the exposed and control subjects, between 2016 and 2019. Differences seem to be consistently greater in the NP-exposed subjects compared with the controls. The selected 14 most differently methylated CG loci were relatively stable in the chronically exposed subjects.	Rossnerova et al., 2021 [96]
TiO_2_, SiNPs and indium tin oxide (ITO)	172 workers from 14 nanomaterial manufacturing and/or handling factories (130 exposed and 43 controls).	Global DNA methylation (HPLC)	Global DNA methylation in white blood cells was decreased in ITO-exposed workers compared with controls.	Liou et al., 2017 [28]

PM: Particulate matter; TiO_2_: Titanium Dioxide; SiNPs: Silica nanoparticles; HPLC: High-performance liquid chromatography.

## Data Availability

Data sharing not applicable.

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
