# Peer review of "The Effect of Nanomaterials on DNA Methylation: A Review"

_nanomaterials, 2023, doi:10.3390/nano13121880_

Round 1

Reviewer 1 Report

In the review manuscript “The effect of nanomaterials on DNA methylation: a review” the authors provide a review of the studies that are available in the literature, with the aim of understanding the potential impact of nanomaterial exposure on DNA methylation and attempting to relate this impact with the nanomaterial’s specific physicochemical characteristics. This topic is of great relevance and this manuscript could provide important reference information that are of considerable interest.

The manuscript is well written and the search strategy appears correct and very clear. The literature review retrieved a total of 181 papers from all databases publications, but only 69 papers were eligible for data analysis and were included in the full text assessment. In addition, the schemes proposed in the figures are appropriate and exhaustive.

I recommend the publication of the manuscript in its present form

Author Response

We are very grateful for your willingness to review our article and we thank you for acknowledging the merit of our work.

Reviewer 2 Report

The authors present a literature review about the existing information about the capability of nanomaterials to induce epigenetic alterations; specifically, DNA methylation. The human information is very limited (only two epidemiological studies with inconsistent results). The remaining in vivo and in vitro studies do not allow to conclude a general trend of the effects of nanomaterials on DNA methylation, although the majority of studies evidenced relevant methylation changes. The authors conclude that epigenetic changes must be considered in the assessment of toxicity induced by nanomaterials.

The review is well planned and developed. The query is logical and transparent, as well as the inclusion/exclusion criteria. To this reviewer’s point of view the main weakness of the review is it seems that the authors have given the same weight to all 69 records that finally passed the different filters. The authors have apparently no assessed the reliability of these 69 papers. It would have been desirable that the records were scored for reliability using some of the established models as ToxRTool or other.

Author Response

The authors are very grateful for your willingness to review this manuscript, for your good appreciation of the work presented, and for your suggestion to improve it. In fact, we believe it is a pertinent suggestion that will be considered in our future works. However, in this case, it would take significantly more time than the 3 days given for the manuscript review to evaluate the 69 articles through this tool, and we apologize for not being able to carry it out. However, taking into account the relevance of suggestion given, we have included the following sentence at the discussion section of the manuscript (lines 324-329), marked as track changes, which we hope can satisfy the reviewer's requirement:

"This review did not intend to be systematic, but rather to summarize the state of the art regarding the knowledge about the potential effects of nanomaterials on DNA methylation. Although all articles considered in this review presented sounded data obtained by well described methods/tools, further refinement of the quality of toxicological data can be achieved using software-based tools, such as “ToxRTool” (Toxicological data Reliability Assessment Tool) [99]".

[99] Schneider et al., 2009, “ToxRTool”, a new tool to assess the reliability of toxicological data, Toxicology Letters, 189(2), 138-144.

Reviewer 3 Report

The manuscript "The effect of nanomaterials on DNA methylation: a review" summarized the studies on the changes in DNA methylation that occur in vivo and in vitro due to treatment by nanomaterials. Whether epigenetic changes occur after nanomaterial exposure is still an open question, and thus the   topic of the manuscript is actual and can be interesting for a wide range of researches. The review includes the analysis of the experimental data and is not just a compilation. The manuscript is well written and well organized, and it is fully within the scope of "Nanomaterials". In my opinion, the review can be published in present form.

Author Response

We are very grateful for your willingness to review our article and are honored by your kind words acknowledging our work.